# DELUCIONQA: Detecting Hallucinations in Domain-specific Question Answering

**Mobashir Sadat**[1*] **Zhengyu Zhou**[2] **Lukas Lange**[2] **Jun Araki**[2] **Arsalan Gundroo**[2]

**Bingqing Wang**[2] **Rakesh R Menon**[3*] **Md Rizwan Parvez**[2] **Zhe Feng**[2]

[1] Computer Science, University of Illinois Chicago

[2] Bosch Research North America & Bosch Center for Artificial Intelligence (BCAI)

[3] UNC Chapel-Hill

msadat3@uic.edu   Zhengyu.Zhou2@us.bosch.com   Lukas.Lange@de.bosch.com

{Jun.Araki, Arsalan.Gundroo, Bingqing.Wang}@us.bosch.com

rrmenon@cs.unc.edu   {Md.Parvez, Zhe.Feng2}@us.bosch.com

## Abstract

Hallucination is a well-known phenomenon in text generated by large language models (LLMs). The existence of hallucinatory responses is found in almost all application scenarios e.g., summarization, question-answering (QA) etc. For applications requiring high reliability (e.g., customer-facing assistants), the potential existence of hallucination in LLM-generated text is a critical problem. The amount of hallucination can be reduced by leveraging information retrieval to provide relevant background information to the LLM. However, LLMs can still generate hallucinatory content for various reasons (e.g., prioritizing its parametric knowledge over the context, failure to capture the relevant information from the context, etc.). Detecting hallucinations through automated methods is thus paramount. To facilitate research in this direction, we introduce a sophisticated dataset, DELUCIONQA[1], that captures hallucinations made by retrieval-augmented LLMs for a domain-specific QA task. Furthermore, we propose a set of hallucination detection methods to serve as baselines for future works from the research community. Analysis and case study are also provided to share valuable insights on hallucination phenomena in the target scenario.

## 1 Introduction

Large Language Models (LLMs) have brought breakthrough performances for many natural language processing tasks, especially for generation-related applications such as chatbots and question-answering (QA). However, state-of-the-art LLMs

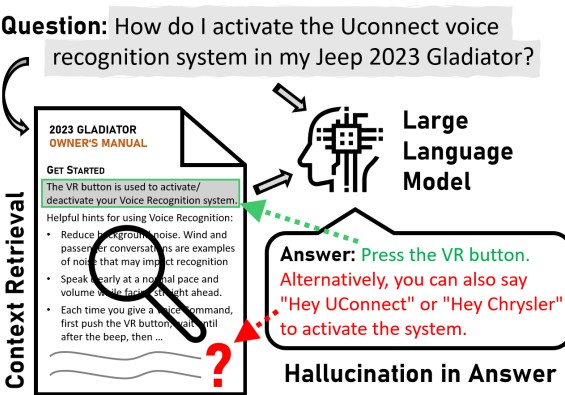

Figure 1: Hallucination in text generated by LLMs.

(e.g., ChatGPT[2], Bard[3]) still have a number of weaknesses. Notably, the tendency to generate hallucinatory (non-factual) statements is a critical issue in LLMs that hinders their usage in applications where reliability is a must-have feature (e.g., vehicle-repair assistant).

To elaborate, hallucination occurs because the models e.g., LLMs follow the principle of maximum likelihood while generating the output text without considering the factuality. As a result, they end up generating outputs that are always highly probable but not necessarily factual. This is a long-standing problem in neural natural language generation and it is pervasive in all types of generation tasks ranging from QA (Su et al., 2022) to abstractive summarization (Falke et al., 2019; Maynez et al., 2020). With the recent technological advances made by state-of-the-art LLMs, hallucination has been largely reduced compared to models from prior work. However, hallucinatory statements are still prevalent in text generated by LLMs.

---

* Work done during an internship at Bosch Research North America.

[1] https://github.com/boschresearch/DelucionQA

[2] https://openai.com/blog/chatgpt

[3] https://bard.google.com/

The potential incorrect/hallucinated content in the answers provided by LLM-based assistants raises significant liability concerns (e.g., vehicle damage) for the applications requiring high reliability.

To date, various approaches have been proposed to reduce the degree of hallucination in QA tasks (Shuster et al., 2021). One effective way is to leverage information retrieval (IR). In particular, instead of directly feeding the question alone to the LLM for answer generation, an IR system first retrieves relevant domain-specific information for the given question to be used as context. Next, the question and the retrieved context are used together in prompt for an LLM to generate an answer. The retrieved context aids the LLM in generating a more reliable answer. However, even with IR and prompt engineering, hallucinations still occur from time to time, typically due to imperfect information retrieval and/or over-reliance on knowledge acquired through pre-training. Detection and mitigation of hallucinatory content is thus an essential topic to widen the application of LLMs to domain-specific QA tasks.

In this work, we focus on addressing hallucination by LLMs in domain-specific QA where high reliability is often needed. Specifically, we present DELUCIONQA, a large dataset for facilitating research on hallucination detection in IR-aided car manual QA systems. We also propose a set of hallucination detection approaches, which may serve as baselines for further improvement from the research community. Our best-performing baseline shows a Macro F1 of only $71.09\%$ on the test set of DELUCIONQA illustrating the difficulty of the task and room for future improvement. Insights on the causes and types of hallucinations for the target scenario are provided in this paper. We hope this work paves the way for more future explorations in this direction, which can expand the scope of LLM usage to tasks with high-reliability requirement.

## 2   Related Work

To date, various resources (e.g., datasets/benchmarks) have been made available for studying hallucination detection. Most of these resources address the existence of hallucinatory content in abstractive summarization and machine translation. For example, Maynez et al. (2020) construct a dataset where each example contains the human annotations for faithfulness and factuality of the abstractive summaries generated by state-of-the-art

models for 500 source texts taken from the XSum dataset (Narayan et al., 2018). Pagnoni et al. (2021) proposes a benchmark named FRANK for understanding factuality in abstractive summarization. The benchmark dataset is derived by generating 2,250 abstractive summaries from different models and then collecting human annotations related to their factuality. Wang et al. (2020) and Kryscinski et al. (2020) also construct smaller scale resources for hallucination in abstractive summarization.

For machine translation, Guerreiro et al. (2023) recently introduced a dataset containing 3,415 examples containing annotations for different types of hallucination. Moreover, there have been significant explorations (Lee et al., 2018; Zhou et al., 2021; Müller and Sennrich, 2021) for addressing hallucination in machine translation.

Recently, several datasets have been proposed to study hallucination in general text generation (not specific to any tasks) by LLMs. Liu et al. (2022) presents HADES, a token-level hallucination detection benchmark for free-form text generation. In contrast to abstractive summarization and machine translation, where the input text can be used as a reference for detecting hallucination, HADES is reference-free. Manakul et al. (2023) released a dataset containing 1,908 sentences with annotations related to hallucination. The sentences are derived by prompting an LLM to generate wikipedia articles about different concepts. Azaria and Mitchell (2023) attempts to automatically detect the truthfulness of statements generated by an LLM by using its activations to train a classifier. To this end, they build a dataset containing true/false human annotated labels from 6 different topics. Rashkin et al. (2023) presents Attributable to Identified Sources or AIS as a framework for evaluating if the text generated by a model is supported by external sources (i.e., not hallucinated).

While all of these datasets are valuable resources for the exploration of hallucination detection by LLMs, none of them cater to customer-facing scenarios where high reliability is essential.

## 3   DELUCIONQA: A Dataset for Hallucination Detection

We aim to develop a dataset that represents a domain-specific QA task requiring high system reliability for customer satisfaction. An example of such a scenario is question answering over car manuals, where incorrect LLM output could cause

damage to cars and endanger lives. However, such domain-specific scenarios pose challenges such as technical terms and language particularities being infrequent during the pre-training stage of LLMs resulting in their sub-optimal representation. To ensure answer reliability and prevent hallucination in these settings, IR technology is usually employed to inject related context information into the LLM prompt. Despite this, hallucinations can still occur.

The DELUCIONQA dataset seeks to capture instances of hallucination in domain-specific QA after using IR, facilitating research on hallucination detection in this scenario. For this, we select car-manual QA as a domain-specific representative task and, without loss of generality, create the dataset using the manual of Jeep's 2023 Gladiator model.

To collect the dataset, we create a set of questions for the technical car repair manual domain and use a QA system equipped with multiple retrievers to generate various retrieval results for each question. We then generate various answers based on those retrieval results, resulting in (question, retrieval result, answer) triples which are annotated by human annotators. The resulting dataset is of high quality and can be used to investigate hallucination detection/handling approaches for domain-specific QA tasks. We elaborate on the dataset curation processes in the following sections.

## 3.1 Question Creation

We create a set of questions for QA on topics related to the Jeep 2023 Gladiator in two steps. First, we use an LLM to generate candidate questions automatically from the domain-specific data resources, in this case the official car manual. For this, we download the publicly available car manual in HTML format and split it into smaller chunks following the HTML paragraph structure. The set of candidate questions is then generated with a multi-task T5 model (Raffel et al., 2020) fine-tuned for question generation and answering.[4] Then, the authors manually filter the set of candidate questions, removing the questions that are not realistic and polishing the remaining questions (e.g., rephrasing a question to what a real user may ask). Additionally, the authors augment the dataset with important questions missing from the automatic generations.

## 3.2 Context Retrieval for QA

For each question (e.g., "how do I activate the Uconnect voice recognition system?"), we retrieve potentially relevant text snippets from the car manual using different IR methods, where the following retrieval methods are evaluated:

**Retrieval Foundation:** We adopt **sparse** and **dense retrieval** functions as foundations for more advanced retrieval methods. For this, we use the Pyserini toolkit (Lin et al., 2021a), which provides reproducible information retrieval results with state-of-the-art retrieval models. The sparse retrieval function is a keyword-based tf-idf variant with Lucene[5]. In contrast, the dense retrieval function is based on FAISS index (Johnson et al., 2019) with a reproduced ColBERT transformer model[6] (Lin et al., 2021b) for embedding and retrieval score.

As the scope of the retrieved context affects the system efficiency as well as the answer quality, we index each document at different granularity level (document-level, section-level, and paragraph-level). This allows the retrieval module to make selections at the appropriate index level and use the context in good scope that fits the question.

**Top-$k$ Ensemble Retrieval:** As the sparse retrieval is keyword-based, it is limited by the exact overlap of tokens and ignores potential synonyms and similar meanings of sentences. Therefore, in addition to the Pyserini approach, we implement an ensemble retrieval function for the QA system, which combines the sparse and dense retrieval methods. The ensemble approach follows a multi-stage retrieval paradigm where we first over-retrieve 5 times the amount of the desired retrieval results. After merging the score from both the dense and sparse retrieval modules, the system reranks and selects the top-$k$ results. In this paper, we experiment with $k = 1$ for the single-best element, and $k = 3$ for a slightly broader retrieval result set, which is to focus on the top-ranked results, and filter out the less relevant content.

**Adaptive Ensemble Retrieval:** It does not guarantee that the static document segmentation at different granularity level would make the returned context to be in best scope that matches the questions. Therefore, the adaptive ensemble retrieval approach is proposed, as an improvement to seek

---

[4] https://huggingface.co/valhalla/t5-small-qg-hl

[5] https://lucene.apache.org
[6] https://huggingface.co/castorini/tct_colbert-v2-hnp-msmarco

for dynamic context range, based on the ensemble retrieval approach introduced above.

The adaptive retrieval checks the retrieval score at different levels, from the top whole-document to the paragraphs at the bottom, which seeks the appropriate scope within a document that matches the input question, and adjusts the retrieval score based on the identified scope. We suppose the document-level score does not sufficiently disclose the precise relevance to the input question. When a low document-level score is calculated, we still dig deeper into the lower level in the structured document, match the relevant text chunks, and eventually combine the relevant chunks, ignoring the less relevant part for the adjusted final retrieval score.

## 3.3 Answer Generation

For generating the answers, we choose the OpenAI ChatGPT model as a representative LLM because it is arguably the most advanced, best performing, and widely accessible LLM at the time of writing this paper. In particular, each retrieval result i.e., the context, is fed together with the question into the OpenAI ChatGPT model with a certain prompt format to generate an answer, resulting in a (Question, Context, Answer) triple. In this work, we use the widely available stable-version gpt-3.5-turbo-0301 model to generate answers. As sometimes different retrieval methods may return the same retrieval result for a question, we further adopt multiple prompting techniques to increase the variety in answers, which will further facilitate the investigation of hallucination. More specifically, for the sparse retrieval result, we construct the prompt with a simple Q+A style prompt. We use a more verbose prompt for the Top-1 and Top-3 ensemble retrieval results. Finally, we use chain-of-thought prompting (Wei et al., 2022) for the adaptive ensemble retrieval and concatenate the search result and the question into a single prompt. More details about the three prompting approaches can be found in Table 6 in the Appendix.

Using the question creation procedure described in Section 3.1, we generate 913 distinct questions that a QA user may ask about the 2023 Jeep Gladiator. Then, we apply the four retrieval methods we study to every question and let ChatGPT generate an answer with the corresponding prompting method. With this, we obtained 3,662 (Question, Context, Answer) triples in total. We then filter out triples with more than 40 context sentences, which

hurts the answer generation, as discovered in our preliminary study. The resultant 2038 examples go through our manual annotation procedure as described next.

## 3.4 Human Annotations

For manually annotating the (Question, Context, Answer) triples constructed from the car-manual QA, we make use of the Amazon Mechanical Turk (MTurk) platform and let crowd-source workers classify each answer sentence if it is supported or conflicted by the given context, or if neither applies when context and answer are unrelated. We also ask the annotators to label whether the question is answerable based on the given context (i.e., if the context is relevant to the question), whether the generated answer contains the information requested in the question, and whether the system conveys an inability to answer the question. The annotation interface can be seen in Appendix B. We take the following steps to ensure high quality of the annotations:

**Test Exercise** Before we present each annotator with the main questionnaire containing a set of three (Question, Context, Answer) triples, we give them a test exercise to complete. The test exercise contains a context and three answer sentences. For each answer sentence, we ask the annotator to detect if it is *supported/conflicted/neither supported nor conflicted* by the context. If the annotator answers all questions correctly for the test exercise, they are moved to the main questionnaire. The test exercise is aimed at evaluating the annotators' understanding of English and their capability to complete the main questionnaire successfully.

**MTurk Filters** We further ensure that the annotations are completed by diligent workers by requiring them to be "Master" turkers with an approval rating of at least 90% and having at least 10,000 tasks approved. In addition, we require the annotators' location to be one of the native English-speaking countries or countries where English is very commonly spoken as a second language.

**Compensation** We set the compensation for the annotators at an hourly rate of $10 which is considerably higher than the US federal minimum wage of $7.25, in order to attract qualified turkers to accept our annotation task.

| Metric | Supported | Neither |
|--------|-----------|---------|
| PRECISION | 97.7% | 89.5% |
| RECALL | 95.6% | 94.4% |
| $F_1$ | 96.6% | 91.9% |

Table 1: Agreement between sentence-level labels based on majority-consensus of annotators and expert-annotated labels. *Conflicted* label is very rare and it does not exist in the subset that was used for this comparison.

## 3.5 The DELUCIONQA Dataset

Based on the annotator responses, we assign a sentence-level label to each sentence in the answer based on a majority vote from the annotators (*supported/conflicted/neither*). In rare cases (for 0.5% of all instances), there was no consensus among the three annotators, which was mainly caused by partially answerable questions according to the given contexts. For these examples, one expert annotator manually inspected the instance and assigned a corrected label.

Based on the sentence-level labels, we then assign the final example-level label for the complete answer. If all sentences in the answer for a certain triple have *supported* labels, the triple is labeled as *Not Hallucinated*. If there exists at least one sentence in the answer that has a *neither* label or a *conflicted* label, we assign a *Hallucinated* label to the triple. In addition, a binary true/false *Answerable* and *Does not answer* label is assigned to each example based on the annotator consensus on whether the context is relevant to the question and whether the LLM refuses to provide an answer (i.e., response implies "I don't know"). Note that if the context is irrelevant (*Answerable* = false) and the LLM does not refuse to provide an answer (*Does not answer* = false), the example is labeled as *Hallucinated*. Therefore, even if there can be cases where the context is irrelevant due to imperfect retrieval methods or the information related to the question being unavailable in the car manual, they do not affect the quality of the labels in our dataset. Our resulting corpus of 2038 examples contains 738 triples labeled as *Hallucinated* and 1300 triples labeled as *Not Hallucinated*.

**Inter-Annotator Agreement** To understand the quality of the labels, we follow two procedures. First, we compute the inter-annotator agreement for the sentence-level labels (*supported/conflicted/neither*) and observe moderate agreement among annotators based on Krippendorff's alpha (55.26) ac-

| Split | #Ques | #Triples | #Hal | #Not Hal |
|-------|-------|----------|------|----------|
| TRAIN | 513 | 1,151 | 392 | 759 |
| DEV | 100 | 216 | 94 | 122 |
| TEST | 300 | 671 | 252 | 419 |
| TOTAL | 913 | 2,038 | 738 | 1,300 |

Table 2: Number of unique questions, number of triples and label distribution in each split of DELUCIONQA. Here, Ques: Question and Hal: Hallucinated.

cording to Landis and Koch (1977). However, there is a high label imbalance in our dataset, as 77% of answer sentences were labeled as supported by the context. This limits the expressiveness of Krippendorff's alpha score, due to its chance adjustment.

Therefore, we compute the agreement between the sentence-level labels based on majority consensus among annotators and sentence-level labels from the annotations performed by one expert annotator for 30 randomly chosen samples. Considering the expert-annotated labels as ground truth, we compute the precision, recall and $F_1$-score for the sentence-level labels and report them in Table 1. Note that no *conflicted* label occurred in this set of samples. As we can see, there is substantial agreement between the expert-annotated labels and the labels from the annotator's majority consensus with 96.6% agreement for *supported* and 91.9% agreement for *neither*. This demonstrates the high quality of our human-labeled and aggregated labels of DELUCIONQA.

**Data Splits** As previously mentioned, we generate multiple triples using different retrieval settings for each question. Thus, the same question can occur multiple times in different triples in our curated dataset. In order to avoid data leakage by having the same question in multiple splits of the dataset (train/development/test), we divide the data at the question level into the usual standard splits. Specifically, out of the 913 unique questions in our dataset, we randomly assigned 513 questions to our training set, 100 to the dev set, and 300 to the test set. After splitting by question, we divide the triples according to their respective split. The number of triples in each split is reported in Table 2. Note that we do not use the training set for the experiments in Section 4, as we explore unsupervised methods. We still provide fixed standard splits to ensure comparability of upcoming research works using this dataset.

**Unanswerable Questions** In addition to the (Question, Context, Answer) triples of DELU-

| Method | Hal | #Exmpl | #Tokens |
|--------|-----|--------|---------|
| Sparse | 39.2% | 590 | 190.9 ± 0.2 |
| Ensemble Top-1 | 28.9% | 574 | 162.3 ± 0.2 |
| Ensemble Top-3 | 23.2% | 258 | 421.2 ± 0.7 |
| A - Ensemble | 45.6% | 616 | 232.5 ± 0.3 |

Table 3: The percentage of Hallucinated examples, total number of examples, and the average ± standard deviations of the number of tokens in the contexts of triples retrieved by each retrieval method. Here, A - Ensemble: Adaptive Ensemble.

CIONQA, we provide a collection of 240 (Question, Context) tuples, for which the LLM refused to provide an answer, e.g., "I don't know the answer" or "There is no mention of *X* in the given context." These instances were filtered before the annotation process using a rule-based method and were not annotated as previously described. Instead, one expert annotator manually checked each question-context pair whether the contexts contained the answer and the LLM failed to answer or if the question was indeed not answerable based on the given context. Out of the total of 240 instances, 20 were identified as answerable. This suggests that the models' abstention to answer can be considered accurate, as it is correct in more than 90% of these cases. We will publish this extra dataset as DELUCIONQA UNANSWERABLE for future studies on this topic. Note that there can still be examples which are not answerable in the main DELUCIONQA dataset (in train/test/dev splits). These examples were undetected by our rule-based method (i.e., they imply "I don't know" in a more subtle manner).

### 3.6 Distribution of Hallucination Cases

We hypothesize that the context retrieved will be more relevant to the question by retrieving with more advanced methods, which meanwhile makes the answer less likely to contain hallucinated content. To test this hypothesis, we divide the triples in our dataset based on the retrieval methods used for their respective context and report the percentage of *Hallucinated* examples for each method in Table 3 on the complete DELUCIONQA dataset.

In general, one can observe that the percentage of hallucination declines with more sophisticated retrieval methods (top 3 methods in Table 3). Interestingly, the contexts retrieved by Adaptive-ensemble search results in the highest percentage of *Hallucinated* answers despite being a stronger method, which highlights the necessity of future re-

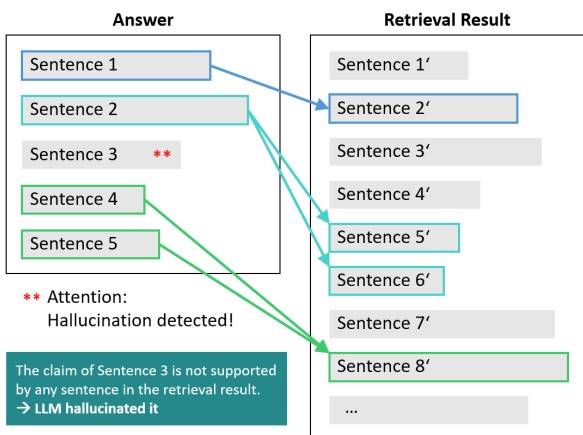

Figure 2: Sentence-similarity based hallucination detection approach.

search on advanced retrieval methods, in particular when working with retrieval-augmented LLMs.

## 4 Automatic Hallucination Detection

This section describes two automatic approaches to detect hallucinations in context-augmented QA. The evaluation of these approaches is reported in Section 5. Note that while we evaluate the proposed hallucination detection approaches using only DELUCIONQA, they can be used for other IR-augmented QA tasks as well.

### 4.1 Sentence-Similarity-based Approach

We first propose an approach based on sentence-level similarity. The basic idea is to determine whether each sentence in the answer is grounded on the retrieval result i.e., the context, as otherwise, the sentences are likely to be hallucinated by the LLM. If there exists one or more sentences that are deemed as not grounded, the answer is predicted to be *Hallucinated*.

The approach calculates two types of similarity measures: sentence-embedding-based similarity and sentence overlapping ratio. We use both measures to detect whether a sentence in the answer is deemed similar to (i.e., grounded on) one or more sentences in the retrieval result, as illustrated in Figure 2, where the third sentence is labeled as hallucinated because it is deemed as not being similar to any sentence in the retrieved context. The two types of similarity calculation evaluate the similarity between two sentences (one in the answer, the other in the retrieval result) from different angles.

The sentence-embedding-based similarity calculates the cosine similarity of the embedding vectors generated by a language model for each of the two

```
for sent_A in answer:
  for sent_R in retrieval_result:
    if embedding_sim(sent_A, sent_R) > T_1):
      similar[sent_A, sent_R] = True
    elif overlapping_ratio(sent_A, sent_R) > T_2:
      similar[sent_A, sent_R] = True
    else:
      similar[sent_A, sent_R] = False
  if not any similar[sent_A, sent_X]
      for sent_X in retrieval_result:
    hallucinated.add(sent_A)
```

Listing 1: Similarity-based hallucination detection that leverages embedding similarity and overlap ratio.

```
keywords = extr(question, retrieval_result, answer)
for k in keywords:
  if k not in retrieval_result:
    hallucinated_k.add(k)
r_hallucinated = len(hallucinated_k) / len(keywords)
if r_hallucinated > T_3:
  hallucination.add(answer)
```

Listing 2: Keyword-based hallucination detection.

sentences in focus. If the cosine similarity score is larger than a certain threshold, the two sentences are deemed similar, that is, the sentence in the answer is predicted to be grounded in the retrieval result.

The second type of similarity is calculated based on the sentence overlapping ratio. It is designed to specifically handle the situation of one-to-many or many-to-one mapping (i.e., one sentence in the answer may merge the information from multiple sentences in the retrieved context, and vice versa.). In this situation, for the focused answer sentence and context sentence, their sentence-embedding similarity score may be low even if the shorter sentence is mostly part of the other sentence, because the longer sentence may contain a significant amount of additional information. The sentence-overlapping-ratio method aims to catch such partial similarity in this situation. The method determines whether two sentences are similar using the following steps:

*Step 1*: Conduct dynamic programming to compare the two sentences with the objective of maximizing overlap, outputting an optimal path that reveals the overlapped words.

*Step 2*: Calculate an overlap length by accumulating the number of tokens for those overlapped phrases that each contains more than either 4 or $n$ words, where $n$ is the 30% of the shorter sentence's length. The relatively short overlap phrases are not considered in order to avoid the influences of noises from common words such as "the".

*Step 3*: Compute the sentence overlap ratio as the overlap length divided by the number of tokens in the shorter sentence. If the ratio is larger than a threshold, the focused two sentences are deemed similar.

These two types of sentence similarity measures have complementary advantages. The embedding-based measure is suitable for a one-to-one mapping, while the overlap-based similarity is advantageous for one-to-many and many-to-one mappings. We further combine them into a hybrid hallucination detection approach, as described in Listing 1. Specifically, if a sentence is deemed as not being similar by the embedding-based measure, the hybrid approach checks if it is deemed as similar by the overlap-based measure. If both measures deem the sentence to be not similar, the sentence is marked as not supported (i.e., hallucinated).

## 4.2 Keyword-Extraction-based Approach

We also propose an approach that leverages keyword extraction (Firoozeh et al., 2020) for detecting hallucination. Here, the main idea of this approach is that given an answer generated by an LLM, if a significant portion of the keywords in the answer does not exist in the retrieved context, the LLM is deemed to have hallucinated the answer. The detailed algorithm of this approach can be seen in Listing 2.

## 4.3 Settings for Hallucination Experiments

In this work, we use the `all-MiniLM-L6-v2`[7] sentence transformer (Reimers and Gurevych, 2019) to calculate the embedding-based similarity between two sentences. Sentence-BERT is chosen here due to its high efficiency and low cost. The keywords for the keyword-extraction-based approach are generated using the OpenAI ChatGPT model (`gpt-3.5-turbo-0301`). The thresholds for all hallucination detection baselines are tuned on the development set of DELUCIONQA using values among $\{0.1, 0.2, ..., 0.9\}$. We found the following thresholds: , $T_1 = 0.6$ for COSINE, $T_2 = 0.1$ for OVERLAP, ($T_1 = 0.1$, $T_2 = 0.9$) for HYBRID and $T_3 = 0.2$ for KEYWORD-MATCH.

---

[7]https://huggingface.co/sentence-transformers/all-MiniLM-L6-v2

| Method | Train | | | Dev | | | Test | | |
|---|---|---|---|---|---|---|---|---|---|
| | Hal | N-Hal | Overall | Hal | N-Hal | Overall | Hal | N-Hal | Overall |
| SIM-COSINE | 63.18 | 74.73 | 70.03 | 72.45 | 77.12 | 74.78 | 63.84 | 73.55 | 69.45 |
| SIM-OVERLAP | 68.47 | 82.72 | 75.59 | 73.51 | 80.16 | 76.84 | 63.89 | 78.28 | 71.09 |
| SIM-HYBRID | 68.73 | 83.16 | 75.94 | 73.51 | 80.16 | 76.84 | 63.33 | 78.29 | 70.81 |
| KEYWORD-MATCH | 30.25 | 77.47 | 53.86 | 31.58 | 69.57 | 50.57 | 31.23 | 74.31 | 52.77 |

Table 4: Class-wise $F_1$ scores (%) and overall Macro $F_1$ scores (%) of the baseline hallucination detection methods on the three splits of DELUCIONQA. Here, Hal: Hallucinated, N-Hal: Not Hallucinated.

## 5 Results and Analyses

This section will report the evaluation of the previously introduced hallucination detection methods on our newly proposed DELUCIONQA dataset.

**Hallucination Detection Performance** We provide a comparison of our hallucination detection algorithms for all three splits of the DELUCIONQA dataset in Table 4. The best-performing model SIM-OVERLAP achieves a Macro $F_1$ of only 71.1% on the unseen test set. This indicates that DELUCIONQA presents a challenging new task with substantial room for future improvement. From Table 4, we also notice that there are fluctuations in performance across the three splits (train/dev/test) of the dataset. Recall that while randomly dividing the data into multiple splits, we ensure that each unique question ends up in a single split (to avoid data leakage). We believe that the observed fluctuations are mainly due to the diversity in questions across the resulting data splits, each containing a disjoint subset of questions.

**Similarity match vs keyword match** The results show that the similarity-based methods (showing Macro F1 scores from $69.45\%$ to $71.09\%$) perform considerably better than the KEYWORD-MATCH approach with $52.8\%$ Macro $F_1$. This indicates that simple keyword matching is not enough for hallucination detection, and more sophisticated methods to capture the similarities between the context and answer are necessary.

**Embedding similarity vs sentence overlap** Comparing the performances of the similarity-based approaches, we can see that SIM-OVERLAP outperforms both of the other methods. This can be mainly attributed to the advantage of SIM-OVERLAP to match answer and context in a one-to-many or many-to-one way, in contrast to SIM-COSINE. That is, if an answer sentence merges the information from multiple sentences from the context (or vice-versa), SIM-COSINE may not be able

to map them because it only calculates the similarity scores at the sentence level. SIM-OVERLAP is better suited to capture these similarities because of its capability in one-to-many mappings. As a result, despite not using any deep semantic embeddings, SIM-OVERLAP performs better than SIM-COSINE. This indicates that DELUCIONQA contains a significant number of one-to-many mappings among the answer and context sentences, and any future methods should incorporate the capability to capture these mappings.

## 6 Case Study and Error Analysis

We observe that hallucination may occur due to various reasons, including insufficient retrieval results, excessive retrieval results, and the LLM focusing on its learned knowledge instead. To facilitate the discussion, we list a number of examples covering various types of hallucination in Table 5.

**P1: Using learned knowledge instead of context.** When the retriever in the QA system fails to fetch the correct information, sometimes the LLM can honestly admit that it does not know the answer, but often the LLM simply makes something up, leading to a hallucinated answer which looks probable. As shown in the first example in Table 5, the system fabricates two non-existing commands, "Hey Uconnect" and "Hey Chrysler", which will never work for a Jeep car. Such commands are hallucinated probably because similar commands like "Hey Google" / "Hi Siri" are frequently found in the LLM's pre-training data.

**P2: Unrelated information in retrieval results.** Even if the retrieved result includes the correct context, the LLM may still hallucinate, likely due to the additional information presented in the retrieval result. For example, for the second question in Table 5, when using the Lucence Search setting, the generated 4-sentence answer is correct except for the last sentence. The last sentence is made up because of excessive information in the retrieval

| Question | Answer |
|---|---|
| How do I activate the UConnect voice recognition system? | ...Alternatively, you can also say **"Hey UConnect"** or **"Hey Chrysler"** to activate the system ...

... or say **"Hey UConnect"** followed by a voice command ... |
| What does emission inspection check? | ...The inspection **also checks for any issues that may cause the MIL to be illuminated during normal vehicle operation** ...

...The inspection also checks for **fluid level**, exhaust system, and **catalytic converter maintenance**. |
| How far can I use key fob to lock/unlock the door? | ...in general, the range can be up to **several hundred feet** ...

... from distances up to 66 ft (20 m) using the key fob ... |

Table 5: Examples of answers generated for given questions, with hallucinated parts in **bold**.

result about unrelated MIL functions. In this case, switching to a better retriever may not guarantee to solve the problem, as shown in the second answer of the question, which is obtained with our Top-3 Ensemble Search. This answer is largely the same, except that it hallucinates the last sentence in a different way due to a different set of excessive information in the retrieval result.

**P1 and P2 can occur jointly.** Note that the LLM does not always hallucinate when the retrieval result includes additional text. Use the third question in Table 5 as an example. While the LLM hallucinates a wrong distance due to insufficient retrieval results when using the Lucence Search setting (see the 1st answer), it successfully answers the question when using the Adaptive Ensemble Search (see the 2nd answer). Although with this setting, the retrieval result contains 25 sentences in total, among which only 2 sentences are related to the answer, the LLM does not hallucinate anything from the large amount of excessive text in retrieval result in this case. This means whether or not an LLM hallucinates also depends on how strongly it is biased towards the pre-learned knowledge.

## 7 Conclusion

This work addresses hallucination in text generated by LLMs in retrieval-augmented question answering, focusing on a domain-specific scenario with high-reliability requirements. Our study demonstrates that even after leveraging IR technology to provide additional context to aid the model in providing a factual answer, hallucinations can still occur due to various reasons. Therefore, it is crucial to develop automatic hallucination detection and handling methods to improve the reliability of LLM-based question-answering systems. To facilitate research in this topic, we release our newly collected DELUCIONQA dataset consisting of 2,038 high-quality human-labeled hallucination ratings for question answering over car manuals. Based on that, we propose a set of initial solutions for hallucination detection as baselines in this paper. Furthermore, we perform a qualitative analysis to provide insights into why hallucination occurs even after the LLM is provided a retrieved-context relevant to the question. While DELUCIONQA is constructed from a single domain (car manual) and a single LLM (ChatGPT), we believe that the insights gained from it (both by us and the research community in the future) will be valuable in addressing hallucinations by other LLMs in the other domains as well. In the future, we will diversify our study of hallucination detection to multiple LLMs and domains, and present more advanced hallucination detection/handling approaches.

## Limitations

While our proposed dataset can facilitate research on the development of methods for both hallucination detection and hallucination handling, our baseline methods are designed only for hallucination detection. Moreover, our dataset exclusively contains answers generated by ChatGPT. At this point, it is not clear whether the methods developed based on our dataset will generalize well for other LLMs.

## Ethics Statement

Considering legal and ethical standards and valuing work effort, we set the compensation for our annotators at the rate of $10/hour, which is above US federal minimum hourly wage, $7.25/hour, at the time of our study.

## Acknowledgements

We thank Paheli Bhattacharya for her contribution in generating the questions that we use to construct our dataset. We also thank our anonymous reviewers for their valuable and constructive feedback, which is helpful to our research work.

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

## A   Prompts

The prompts for generating the answers given the question and the context from each triple can be seen in Table 6. For extracting the keywords from the answer in our keyword based hallucination detection method, we use the following prompt: *<answer> Given the paragraph above, show the keywords in it:*.

## B   Annotation User Interface

We can see the user interface shown to the annotators. In Figure 3, we can see the test exercise that every annotator need to pass to be able to move to the main questions. This exercise is set up to ensure that the annotators have a good command over English and are qualified to complete the task. The main questionnaire is shown in two parts in Figure 4 and 5. The left side of the main questionnaire shows the question, context and the answer sentences from each triple. The questions are then asked in the right side.

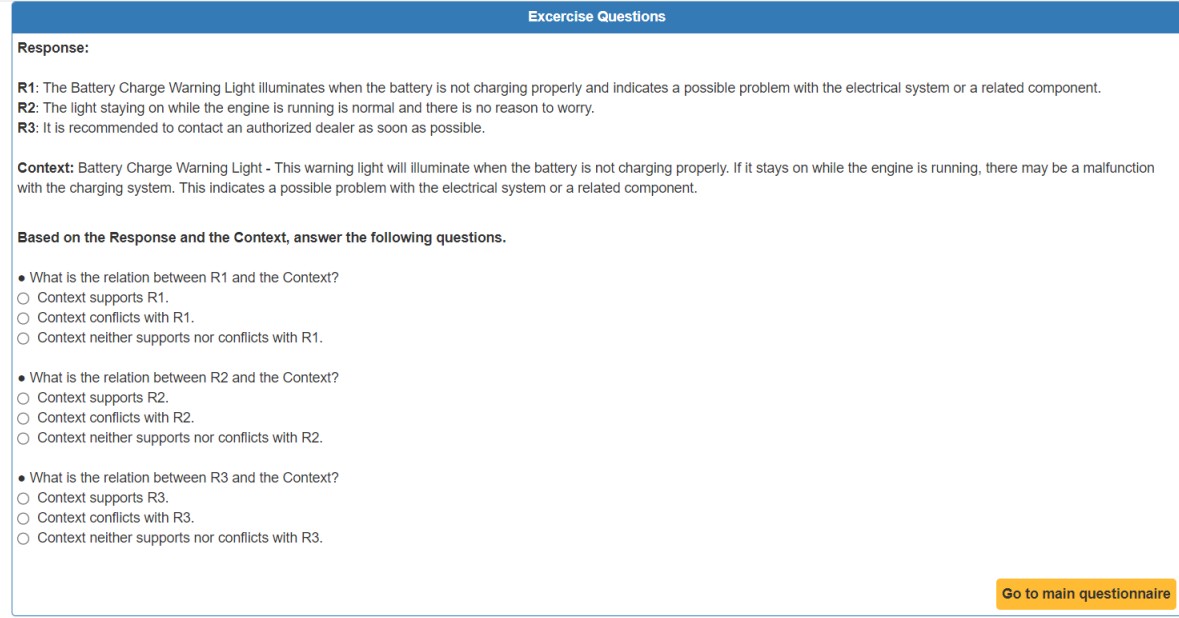

Figure 3: Test exercise for crowd-workers.

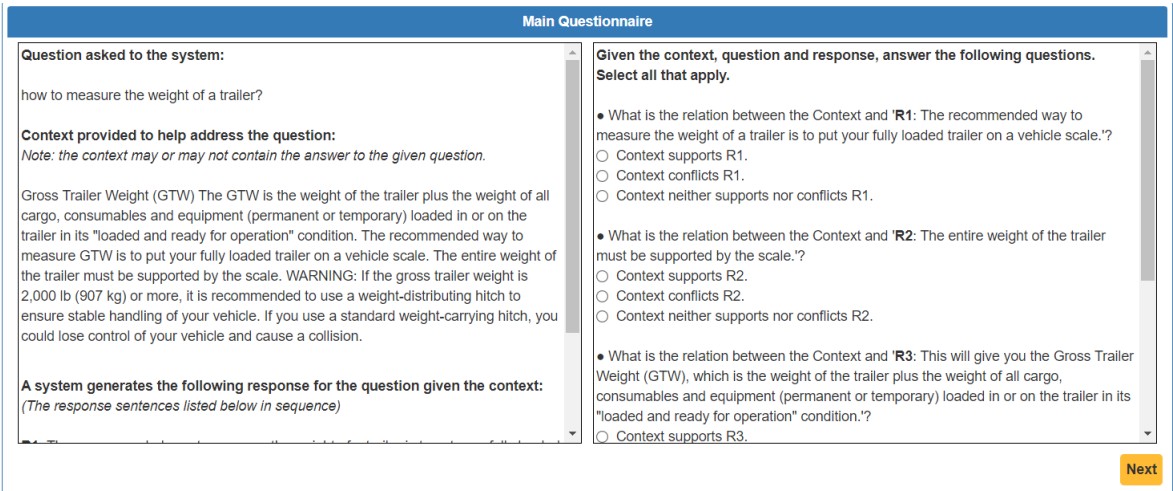

Figure 4: Main Questionnaire Part 1.

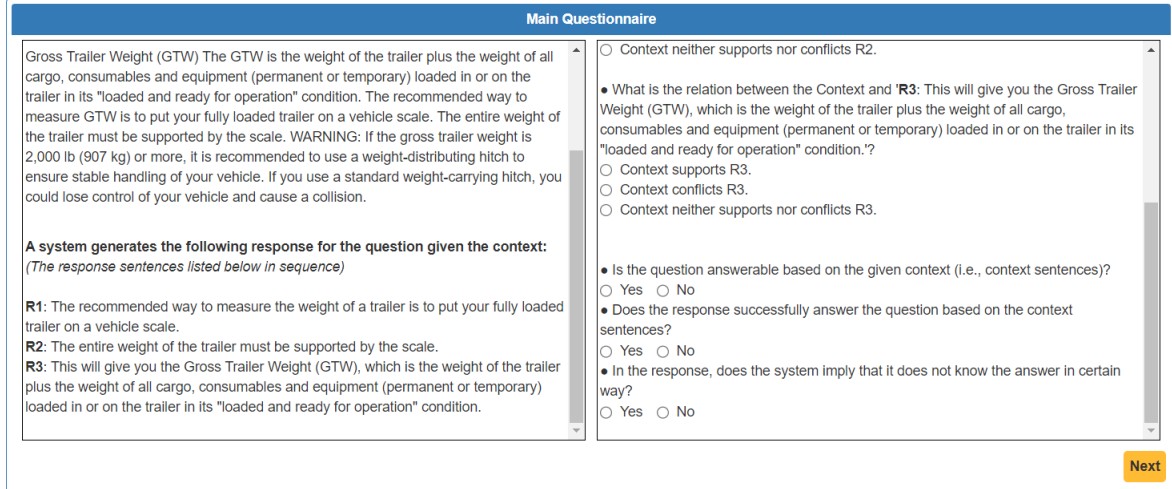

Figure 5: Main Questionnaire Part 2.

| Retrieval Method | Prompt Type | Prompt |
|---|---|---|
| Sparse | Q+A | {"role": "system", "content": "You are a helpful and kind AI Assistant."}
{"role": "user", "content": <context> + "\n\n" + "Q: " + <question> + "\n" + "A:"} |
| Ensemble Top-1 | Verbose | {"role": "system", "content": "You are a helpful and kind AI Assistant."}
{"role": "user", "content": "Given the context: \n" + <context> + "\n\n" + "Please answer the question:\n" + <question>"} |
| Ensemble Top-3 | Verbose | {"role": "system", "content": "You are a helpful and kind AI Assistant."}
{"role": "user", "content": "Given the context: \n" + <context> + "\n\n" + "Please answer the question:\n" + <question>"} |
| Adaptive Ensemble | COT. | {"role": "system", "content": "You are a helpful and kind AI Assistant. If there is multiple answer/reasons depending on the situation you can ask for my situation and provide answers/reasons step-by-step as a list."}
{"role": "user", "content": <context> + "\n\n" + "Q: " + <question> + "\n" + "A:"} |

Table 6: The prompts used to generated answers using the question and context generated with different retrieval methods. Here, COT stands for Chain-of-thought. <context> and <question> are replaced by actual context and question of each example.