# OpenReview forum: "DelucionQA: Detecting Hallucinations in Domain-specific Question Answering"
_EMNLP/2023/Conference — EMNLP 2023 Findings_

### Official Review · Reviewer_Q4DD · 2023-07-28

**Typos Grammar Style And Presentation Improvements:** 1. Line 285 "aplly"
**Soundness:** 4

**Excitement:**

3: Ambivalent: It has merits (e.g., it reports state-of-the-art results, the idea is nice), but there are key weaknesses (e.g., it describes incremental work), and it can significantly benefit from another round of revision. However, I won't object to accepting it if my co-reviewers champion it.

**Paper Topic And Main Contributions:**

The paper proposed DelucionQA, a QA dataset based on manual of Jeep’s 2023 Gladiator model, and evaluated the sentnece level hallucination of the ChatGPT generated answer. The paper also provides two baselines for automaticly hallucination evaluation.
The main contributions:
1. The paper proposed the DelucionQA, a QA dataset that provides (Question, Context, and ChatGPT generated answer).
2. The paper did a sentence level human evaluation of hallucination for the ChatGPT generated answer, which can boost further analysis of the  LLMs' hallucination.
3. The paper introduce several baseline approaches (Sentence-Similarity-based Approach, Keyword-Extraction-based Approach) for hallucination detection.

**Questions For The Authors:**

1. In Table 4 results. Why the F1 socres for the same methods are so different among different data split (train val trest)? Does it mean the dataset is not consist enough among different split?
2. There are many domain specific QA datasets that provide context, question and answer, especially in the medical domain (such as PubMedQA). What's the difference between them and this work? They even provide the gold answer rather than the ChatGPT answer.

**Reasons To Accept:**

1. The paper is well-written and easy to understand.
2. The proposed dataset is carefully designed and annoted by human.
3.  The proposed baseline for hallucination detection approaches are partially effective (Similarity based approaches achieved a overall performance higher than 70% ).
4.

**Reasons To Reject:**

1. The Context for each question is retrieved automatically without manual verification, which could lead some errors in the dataset (all context of a specific question is not relevant to the question, The generated answer is correct but can not verified by the context because the context does not provide the all relevant information)
2. The dataset does not provide a gold answer for the question, which could limited the use case of the proposed dataset.

**Reproducibility:**

4: Could mostly reproduce the results, but there may be some variation because of sample variance or minor variations in their interpretation of the protocol or method.

**Reviewer Confidence:**

4: Quite sure. I tried to check the important points carefully. It's unlikely, though conceivable, that I missed something that should affect my ratings.

---

> ### Author Rebuttal · Authors · 2023-08-29
>
> Thank you for reviewing our paper. We address your concerns below.
>
> >> Regarding relevance of the retrieved context
> (Reviewer comment: "The Context for each question is retrieved automatically without manual verification,...")
>
> Indeed, our information retrieval (IR) module can retrieve context that is irrelevant or only partially relevant to the question. Moreover, it is also possible that the model generates a factual answer despite being given an irrelevant context. However, in this study, we focus on hallucination phenomena, especially for real-world domain-specific QA equipped with state-of-the-art retrieval  methods and state-of-the-art LLM.  Thus, instead of labeling correct/wrong, we define/label hallucination as whether all content in an answer is supported by the context. If the context is irrelevant, the expected behavior is that the model generates an answer that implies “I don’t know.” Therefore, even though the dataset potentially contains examples where the context is irrelevant or partially relevant, they do not lead to erroneous labels.
>
> >> Regarding gold answer for the question
> (Reviewer comment: "The dataset does not provide a gold answer for the question,...")
>
> The objective in building our dataset is to facilitate the research on detecting hallucinations occurring in answers generated by retrieval-augmented LLMs. While having ground truth answers can result in additional use cases of the dataset, we believe they are unnecessary for hallucination investigation/detection.
>
> >> Regarding the question on F1 scores being different for different data splits
> (Reviewer question: "In Table 4 results. Why the F1 socres for the same methods are so different among different data split...")
>
> We follow the same annotation strategy for all <question, context, answer> triples in our dataset. After we obtain the hallucination labels for all examples, we randomly divide the dataset into the three splits at the question level to avoid any data leakage by having the same question in multiple splits. In this way, the characteristics of the resulting splits should be consistent. Nevertheless, as we report in Table 2, distribution of the Hallucinated vs. Not Hallucinated classes in train, test and dev splits end up different from each other, due to the diversity of the questions across the splits. For example, the percentage of hallucinated examples in the dev set is 43.51% whereas it is 34.40% and 37.70% in the train and test sets, respectively. This difference in class distribution could be a reason behind the fluctuations of the performance of the proposed methods. Also, the size of our dataset is relatively small compared to other tasks (e.g., those used in sentiment analysis) and the proposed approaches only tune a small number of hyperparameters. These also contribute to the observed fluctuations on the performance of the proposed methods.
>
> >> Regarding the question on the difference between our dataset and other domain-specific QA datasets
> (Reviewer question: "There are many domain specific QA datasets that provide context, question and answer...")
>
> The task in the existing QA datasets (e.g., PubMedQA) is to generate the answer based on the context and the question. As a result, the answer in each example is written by humans (i.e., gold answer). In contrast, the task presented by our dataset is to detect hallucination in the answers generated by LLMs. As such, the answers in our dataset are not gold answers (not written by humans). Rather, the hallucination labels (Hallucinated or Not Hallucinated) are gold labels assigned based on human annotation. We will make sure to clearly explain the differences between the existing QA datasets and our dataset in the paper.
>
> >> Presentation Improvements
>
> Thank you for pointing out the typo. We will update the paper accordingly.

---

### Official Review · Reviewer_1Jdn · 2023-07-30

**Paper Topic And Main Contributions:** 1. introduce a dataset, DelucionQA, t…
**Soundness:** 3

**Excitement:**

3: Ambivalent: It has merits (e.g., it reports state-of-the-art results, the idea is nice), but there are key weaknesses (e.g., it describes incremental work), and it can significantly benefit from another round of revision. However, I won't object to accepting it if my co-reviewers champion it.

**Missing References:**

1. Rashkin, Hannah, et al. "Measuring attribution in natural language generation models." Computational Linguistics (2023): 1-66.

2. Ji, Ziwei, et al. "Survey of hallucination in natural language generation." ACM Computing Surveys 55.12 (2023): 1-38.

**Questions For The Authors:**

In Figure 2 What if sentence3 is the summary of context or sentence3 entails the context, even though it is not sentence-similar to them?

**Reasons To Accept:**

1. Hallucination in LLM is getting more and more attention and this problem

2. The dataset caters to customer-facing scenarios where high reliability is essential.

**Reasons To Reject:**

1. The method to build a dataset is not novel.

2. I don’t think it’s reasonable that use sentence similarity between each sentence of generation and context to detect hallucination.

**Reproducibility:**

4: Could mostly reproduce the results, but there may be some variation because of sample variance or minor variations in their interpretation of the protocol or method.

**Reviewer Confidence:**

4: Quite sure. I tried to check the important points carefully. It's unlikely, though conceivable, that I missed something that should affect my ratings.

---

> ### Author Rebuttal · Authors · 2023-08-29
>
> Thank you for your feedback. We address your concerns below.
>
> >> Novelty of the method for building the dataset
> (Reviewer comment: "The method to build a dataset is not novel.")
>
> We would like to note that the novelty in our paper comes more from the dataset itself, rather than its construction method. More specifically, the goal of this paper is to provide a dataset to investigate hallucination for retrieval-based QA and the corresponding hallucination detection methodologies. To our knowledge, there are no existing datasets that contain <retrieved context, question, answer> triples with their manually annotated hallucination labels. In addition to the hallucination labels, we will also release the “Answerability” labels indicating whether each question is answerable based on the given context. We believe that our dataset will be a valuable resource for the advance of research in the topic of hallucination detection.
>
> >> Regarding using sentence similarity for hallucination detection
> (Reviewer comment: "I don’t think it’s reasonable that use sentence similarity...")
>
> The proposed sentence-similarity based hallucination detection approaches aim to detect those answer sentences that are not grounded on the retrieved context, that is, being hallucinated.  The approaches determine whether or not an answer sentence is grounded on one or more context sentences based on similarity (i.e., semantic similarity, partial token similarity).  The SIM-COSINE approach attempts to capture the semantic similarity between one answer sentence and one context sentence by measuring the cosine similarity in the embedding space, the SIM-OVERLAP method attempts to capture the one-to-many or many-to-one mapping between answer and context sentences based on the ratio of partial token overlap, and their hybrid (i.e., SIM-HYBRID) attempts to combine the complementary benefits of the two approaches.  Although these approaches do not cover many corner cases, they can handle a significant portion of hallucinations, achieving ~70% macro F1 on testing data. Therefore, they are suitable to serve as solid baselines for future research on hallucination detection for retrieval-based QA.
>
> >> Regarding the question “Figure 2 What if sentence3 is the summary of context or sentence3 entails the context, even though it is not sentence-similar to them?”
>
> If an answer sentence (e.g.,sentence 3) is the summary of context or entails the context, it will be determined as similar to one of the context sentences if the partial token overlap ratio (i.e., overlapping_ratio()) is larger than the threshold T_2 or if their embedding similarity (i.e., embedding_sim()) is larger than the threshold T_1. Otherwise, if it is determined as dissimilar to any context sentence, it will be labeled as hallucinated.
>
> If two sentences (one from context and the other from answer) are dissimilar at the surface-level (i.e., low overlap of tokens) despite being supported by each other, the token overlap (SIM-OVERLAP) based method will fail to make the correct prediction. However, if these two sentences are indeed supported by each other, they will be semantically similar and therefore, they will be close to each other in the embedding space despite a possible low overlap of tokens. Since our SIM-COSINE method measures the similarity based on the closeness of sentences in the embedding space, it should be able to capture the fact that these two sentences are semantically similar and make the correct prediction.
>
> Nevertheless, as discussed previously, the hallucination detection approaches we proposed to serve as preliminary baseline methods do not well cover corner cases yet.  Thus, one future research direction of hallucination detection can be the reduction of such missed corner cases.
>
> >> Regarding Missing References
>
> Thank you for pointing out the missing references. We will update the paper accordingly.

---

### Official Review · Reviewer_6jrh · 2023-08-05

**Typos Grammar Style And Presentation Improvements:** Line 285 - “we aplly” —> “we apply”
**Soundness:** 3

**Excitement:**

3: Ambivalent: It has merits (e.g., it reports state-of-the-art results, the idea is nice), but there are key weaknesses (e.g., it describes incremental work), and it can significantly benefit from another round of revision. However, I won't object to accepting it if my co-reviewers champion it.

**Paper Topic And Main Contributions:**

This paper focuses on the problem of hallucination in the car manual-based question-answering system. Authors created a new dataset (DelucionQA) by i) creating diverse questions based on the car manual, ii) running context retrieval for the created Q (explores 3 different retrieval models), and iii) using the ChatGPT model (gpt-3.5-turbo) to generate answers. Human annotation is collected for each triple (question, answer, context) to see if the answer is hallucinated or not. Using the collected dataset, the authors describe two automatic ways of detecting hallucination and provide some analysis of how hallucination manifests in ChatGPT responses.



**Reasons To Accept:**

1. Hallucination is an important and timely problem that needs to be addressed.
2. Collected dataset can be helpful for understanding the hallucinating behaviour of ChatGPT better, and can be used to also train a hallucination detection model to serve as an automatic metric for further work


**Reasons To Reject:**

1. The insights from this paper are rather shallow due to the following reasons: 1) restricted to car-manual qa setting, 2) restricted to one model (ChatGPT), 3) “LLM using learned knowledge instead of context” is an insight that has been already discovered in multiple pieces of literature. “Unrelated information in retrieval results” causing LLM is also a rather expected behaviour. If given the wrong context as prompt/input, LLMs are expected to also have errors propagated..
2. Another potential usage of the collected dataset would be to: train a hallucination detector. However, since this dataset only has generations from only one LLM, it is unclear if the hallucination detector trained on this dataset will generalize to other LLMs.
3. The proposed automatic hallucination detection methods (i.e., sentence-similarity-based approach and keyword-extraction-based approach) are not novel, and I'm not convinced that these are reliable ways to do so.

**Reproducibility:**

3: Could reproduce the results with some difficulty. The settings of parameters are underspecified or subjectively determined; the training/evaluation data are not widely available.

**Reviewer Confidence:**

4: Quite sure. I tried to check the important points carefully. It's unlikely, though conceivable, that I missed something that should affect my ratings.

---

> ### Author Rebuttal · Authors · 2023-08-29
>
> Thank you for reviewing our paper and your feedback. We address your concerns below.
>
> >> Regarding limiting the dataset to car-manual QA and a single LLM
> (Reviewer comment: "The insights from this paper are rather shallow due to the following reasons: 1) restricted to car-manual qa setting, 2) restricted to one model (ChatGPT), 3)... ")
>
> We construct a dataset that facilitates research in the topic of hallucination detection in retrieval-augmented question answering for domains where high reliability is required. As one might expect, manually annotating labels for this task is quite time consuming and therefore, expensive. As such, we chose car-manual QA, which is a grounded real-world setting where erroneous answers may potentially result in hazardous outcome, as a representative domain and chose ChatGPT, which was arguably the most advanced, best performing and widely accessible LLM when we conducted the experiments (GPT-4 was not available at that time), as a representative model to generate the dataset. To our knowledge, there is no other existing dataset that contains the hallucination detection labels for retrieval augmented question answering. While our dataset is indeed constructed from a single domain and single LLM, the insights gained from it (both by us in the paper and in the future by the research community) will be valuable in addressing hallucinations by other LLMs in the other domains as well. Moreover, the hallucination detection methodologies proposed to serve as baselines for future research, which will be discussed more below, are independent of the car-manual QA and ChatGPT, being generally applicable for retrieval-augmented QA. Nevertheless, we may further extend the dataset to other LLM and/or domains in the future.
>
> >> Regarding the insights in the paper being shallow
> (Reviewer comment: "The insights from this paper are rather shallow due to the following reasons: 1) …, 2) …, 3) “LLM using learned knowledge instead of context” is an insight that has been already discovered..." )
>
> As hallucination is not a new topic for LLMs, it is natural that some observations reported in this paper have already been mentioned in multiple previous works. However, one unique contribution of this paper is the provision of a dataset that enables systematic investigation of hallucination phenomena especially for retrieval-based QA, which is an essential task to enable safe application of LLMs to applications with high reliability requirements. Using the collected dataset, we provide a systematic overview of the occurrence of hallucinations for retrieval-based QA, which has not been covered in previous literature. In addition to those observations that confirm the findings reported in previous papers, the additional insights as well as the systematic view we provided are also valuable for researchers working in this area. For example, we point out that "unrelated information in retrieval results" may or may NOT result in hallucination in a generated answer, with concrete examples given in the case study.  We also point out that hallucinations also depend on how strongly the LLM is biased towards the pre-learned knowledge. The relation between the bias of pre-learned knowledge and the retrieved context could be a meaningful research direction to identify the reason for hallucination. We will make this clear in the revised version.
>
>
> >> Regarding the generalizability of hallucination detector trained on our dataset
> (Reviewer comment: "Another potential usage of the collected dataset would be to: train a hallucination detector...")
>
> Kindly note that the goal of this paper is to provide a dataset to investigate hallucination for retrieval-based QA and the corresponding hallucination detection methodologies. We provide a set of hallucination detection approaches to serve as baselines for future hallucination detection research, which are designed to be generally applicable for retrieval-based QA, algorithm-wise being independent of the LLM used to generate the answer. If another LLM is used and/or another domain (not car-manual) is adopted for retrieval-based QA, the proposed approaches can be applied to the new application simply by tuning thresholds (i.e., T_1, T_2, T_3 in Listing 1&2) on the new development dataset. The generalizability of a hallucination detector trained on the collected dataset to other LLMs is out of the scope of this paper. Though it is an interesting topic that we will investigate in our future work.
>
> >> Regarding the novelty and reliability of the proposed hallucination detection methods
> (Reviewer comment: "The proposed automatic hallucination detection methods (i.e., sentence-similarity-based approach and keyword-extraction-based approach) are not novel,...")
>
> The hallucination detection methods proposed in this paper are especially designed for retrieval-based QA with LLMs. To our knowledge, there has not been any prior algorithm reported in literature for this area.  If the reviewer could point out relevant literature(s) that raised the novelty concern, we would be happy to acknowledge and/or discuss the differences further.
>
> Our proposed methods (i.e., sentence-similarity-based approach and keyword-extraction-based approach) are independently developed for the focused area, i.e., retrieval-based QA with LLMs, based on two valid assumptions, respectively: 1) if an answer sentence is similar to one or more sentences in the retrieved context, then this answer sentence is not hallucinated, and 2) if a large portion of keywords in an answer do not show up in the retrieved context, then the whole answer is likely to be hallucinated. The developed algorithms are intuitive and honestly reflect the two assumptions. The experimental results provide valuable insights regarding these two assumptions on hallucination detection. And the sentence-similarity-based approaches, which do not require supervised training at all, achieve  ~70% macro F1 on testing data. This shows that the proposed approaches are solid and serve well as baselines for future hallucination-detection research.
>
> >> Presentation Improvements
>
> Thank you for pointing out the typo. We will update the paper accordingly.

---

### Meta-Review · Area_Chair_8TJQ · 2023-09-04

**Recommendation:** 3
**Confidence:** 3
**Best Paper Recommendation:** No

**Metareview:**

This paper focuses on hallucination detection in the domain of car manual-based QA. The paper introduces a new dataset, DelucionQA, for this task based on the manual of Jeep's 2023 Gladiator model. They use ChatGPT to generate answers and manually verified hallucination in answers based on the question, answer, and context. The authors then use the proposed dataset to develop hallucination detection systems.

Pros:
1. They introduce a new dataset for hallucination detection.
2. The paper benchmarks the proposed dataset with several methods.

Cons:
1. Limited generalizability. The data is only based on the manual of Jeep's 2023 Gladiator model and answers from the ChatGPT model. Reviewers point out that it is unclear how well the proposed dataset can generalize to different domains and LLMs (Language Model Models).
2. Lack of reliability in hallucination detection models. Reviewers point out that it is a reasonable approach to detect hallucination by measuring the similarity between the model's answer and context. More sophisticated methods and analyses are expected.

**Meta-Review:**

The paper focuses on an important and interesting topic: LLM hallucination detection.
The contributions and scope of this paper are limited. The proposed method also needs more justifications.

---

### Decision · Program_Chairs · 2023-10-07

**Decision:**

Accept-Findings

**Comment:**

This paper focuses on hallucination detection in the domain of car manual-based QA. The paper introduces a new dataset, DelucionQA, for this task based on the manual of Jeep's 2023 Gladiator model. They use ChatGPT to generate answers and manually verified hallucination in answers based on the question, answer, and context. The authors then use the proposed dataset to develop hallucination detection systems.

Pros:
1. They introduce a new dataset for hallucination detection.
2. The paper benchmarks the proposed dataset with several methods.

Cons:
1. Limited generalizability. The data is only based on the manual of Jeep's 2023 Gladiator model and answers from the ChatGPT model. Reviewers point out that it is unclear how well the proposed dataset can generalize to different domains and LLMs (Language Model Models).
2. Lack of reliability in hallucination detection models. Reviewers point out that it is a reasonable approach to detect hallucination by measuring the similarity between the model's answer and context. More sophisticated methods and analyses are expected.